# Newborns from Mothers Who Intensely Consumed Sucralose during Pregnancy Are Heavier and Exhibit Markers of Metabolic Alteration and Low-Grade Systemic Inflammation: A Cross-Sectional, Prospective Study

**DOI:** 10.3390/biomedicines11030650

**Published:** 2023-02-21

**Authors:** José Alfredo Aguayo-Guerrero, Lucía Angélica Méndez-García, Aarón Noe Manjarrez-Reyna, Marcela Esquivel-Velázquez, Sonia León-Cabrera, Guillermo Meléndez, Elena Zambrano, Espiridión Ramos-Martínez, José Manuel Fragoso, Juan Carlos Briones-Garduño, Galileo Escobedo

**Affiliations:** 1Laboratory of Immunometabolism, Research Division, General Hospital of Mexico “Dr. Eduardo Liceaga”, Mexico City 06720, Mexico; 2Laboratory of Proteomics, Research Division, General Hospital of Mexico “Dr. Eduardo Liceaga”, Mexico City 06720, Mexico; 3Unidad de Biomedicina, Facultad de Estudios Superiores-Iztacala, Universidad Nacional Autónoma de México, Tlalnepantla 54090, Mexico; 4Carrera de Médico Cirujano, Facultad de Estudios Superiores-Iztacala, Universidad Nacional Autónoma de México, Tlalnepantla 54090, Mexico; 5Research Department, Universidad Autónoma de Nuevo León, Monterrey 64460, Mexico; 6Reproductive Biology Department, Instituto Nacional de Ciencias Médicas y Nutrición Salvador Zubirán, Mexico City 14080, Mexico; 7Experimental Medicine Research Unit, Facultad de Medicina, Universidad Nacional Autónoma de México, Mexico City 06720, Mexico; 8Department of Molecular Biology, Instituto Nacional de Cardiología “Ignacio Chávez”, Juan Badiano 1, Sección XVI, Tlalpan, Mexico City 14080, Mexico; 9Department of Gynecology, General Hospital of Mexico “Dr. Eduardo Liceaga”, Mexico City 06720, Mexico

**Keywords:** sucralose, pregnancy, neonate, birth weight, cord blood, insulin, monocyte subset, TNF-alpha, IL-10

## Abstract

Robust data in animals show that sucralose intake during gestation can predispose the offspring to weight gain, metabolic disturbances, and low-grade systemic inflammation; however, concluding information remains elusive in humans. In this cross-sectional, prospective study, we examined the birth weight, glucose and insulin cord blood levels, monocyte subsets, and inflammatory cytokine profile in 292 neonates at term from mothers with light sucralose ingestion (LSI) of less than 60 mg sucralose/week or heavy sucralose intake (HSI) of more than 36 mg sucralose/day during pregnancy. Mothers in the LSI (*n* = 205) or HSI (*n* = 87) groups showed no differences in age, pregestational body mass index, blood pressure, and glucose tolerance. Although there were no differences in glucose, infants from HSI mothers displayed significant increases in birth weight and insulin compared to newborns from LSI mothers. Newborns from HSI mothers showed a substantial increase in the percentage of inflammatory nonclassical monocytes compared to neonates from LSI mothers. Umbilical cord tissue of infants from HSI mothers exhibited higher IL-1 beta and TNF-alpha with lower IL-10 expression than that found in newborns from LSI mothers. Present results demonstrate that heavy sucralose ingestion during pregnancy affects neonates’ anthropometric, metabolic, and inflammatory features.

## 1. Introduction

The global burden of obesity and type 2 diabetes (T2D) has dramatically increased worldwide in the last 30 years [1,2]. In the US, the prevalence of overweight and obesity grades 1, 2, and 3 substantially increased from 2000 to 2010, where 65.5% of women and 71.1% of men showed an abnormally high body mass index (BMI) [3]. In the same period, the consumption of non-nutritive sweeteners (NNS) raised 54% in adults and 200% in children from the US, especially sucralose [4]. Sucralose is the bestselling NNS among millions of global purchasers who demand sweet-tasting food and beverages with reduced calories and sugar [5,6,7]. Sucralose is also one of the most ingested NNS, especially by women of reproductive age who aim to avoid excessive weight gain during pregnancy [8,9]. Although the Food and Drug Administration (FDA) considers sucralose safe for pregnant women to consume, recent evidence indicates that sucralose ingestion during pregnancy may increase the offspring’s susceptibility to weight gain [10,11].

Cai et al. conducted a meta-analysis finding that women who consumed sucralose and other NNS during pregnancy delivered babies with an increased birth weight and a decreased gestational age compared to women reporting not to utilize NNS [12]. Moreover, the ingestion of aspartame or sucralose during gestation increased body mass index (BMI) and fat mass of children at three years, while also producing elevated body weight and adiposity in C57BL/6 mice [13]. This information supports the role of sucralose intake during pregnancy in increasing the progeny’s risk of developing obesity early in life. However, excessive weight gain does not come alone because it is often accompanied by metabolic disturbances and low-grade systemic inflammation, which actively contribute to the onset of chronic non-communicable diseases such as T2D.

Azad et al. found that exposing female mice to sucralose during pregnancy and lactation increases body weight, adipocyte hypertrophy, glucose intolerance, and insulin resistance in the offspring at 11 weeks of age [13]. In parallel, feeding pregnant mice with sucralose magnifies high-fat diet (HFD)-induced liver steatosis while amplifying tumor necrosis factor-alpha (TNF-alpha) expression in the progeny’s hepatic tissue at 12 weeks of age [14]. Furthermore, our research team recently demonstrated that sucralose consumption increases serum insulin levels and alters the monocyte subset balance in young women, with particular emphasis on classical and nonclassical monocytes [15]. Together with high interleukin (IL)-1 beta expression and low IL-10 mRNA levels, TNF-alpha expression and monocyte subset alteration are well-known molecular markers of low-grade systemic inflammation observed in patients with obesity and metabolic syndrome [16,17].

Even though robust data in animal models show that sucralose intake during gestation can predispose the offspring to weight gain, metabolic disturbances, and low-grade systemic inflammation, concluding information remains elusive in humans. Herein, we examined the effects of sucralose ingestion during pregnancy on markers of metabolic dysfunction and systemic inflammation, including birth weight, glucose and insulin levels, monocyte subsets; and IL-1 beta, TNF-alpha, and IL-10 expression in a large cohort of neonates born at term.

## 2. Materials and Methods

### 2.1. Participants

We conducted a cross-sectional, prospective, observational study, enrolling 292 neonates born at term from mothers aged 18–30 with low-risk pregnancies who gave birth by spontaneous vaginal delivery between 37 and 41 weeks. Pregnant women enrolled in the study voluntarily attended the Department of Gynecology of the General Hospital of Mexico to start prenatal care at the beginning of the second trimester of pregnancy. The enrollment of participants took place from April 2018 to March 2020 and August 2021 to September 2022. All the pregnant women enrolled in the study provided written informed consent, and fathers provided written informed assent, agreeing for their infants to participate in the research that the Institutional Ethical Committee of the General Hospital of Mexico had previously approved. We excluded pregnant women from the study with any pregnancy complications, including gestational diabetes, preeclampsia, eclampsia, placental complications, ectopic pregnancy, bleeding, or amniotic fluid complications. Moreover, we excluded pregnant women with a previous diagnosis of T2D, hypertension, endocrine disorders, infectious diseases, chronic inflammatory illnesses, autoimmune disease, human immunodeficiency virus (HIV) seropositivity, hepatitis C virus (HCV) seropositivity, or hepatitis B virus (HVB) seropositivity; and women under any anti-inflammatory or immunomodulatory drug therapy in the last six months. We also excluded pregnant women if they had more than 14 weeks of pregnancy at the time of the study’s beginning to avoid missing information regarding consuming sucralose or any other NNS during the last two weeks of pregnancy. We eliminated from the study participants previously enrolled who refused to keep participating or from whom we could not obtain all the demographic, anthropometric, clinical, and biochemical data. We conducted all the minimally invasive procedures in mothers and neonates in strict adherence to the principles of the 1964 Declaration of Helsinki and its posterior amendment in 2013, under the supervision of the Institutional Ethical Committee of the General Hospital of Mexico with the registration of the project number DI/17/UME/03/090.

### 2.2. Demographic, Anthropometric, and Clinical Measurements

We registered relevant demographic, anthropometric, and clinical data in all the pregnant women enrolled in the study using the digital version of the electronic records of the General Hospital of Mexico (ERGHM). The demographic, anthropometric, and clinical data obtained by ERGHM included age, pre-gestational BMI, number of previous pregnancies, pregnancy age, and blood pressure. We performed oral glucose tolerance tests (OGTT) in all the pregnant women at the beginning of the third trimester of pregnancy, starting with a 75 g glucose load at 0 min, measuring blood glucose levels at 0, 30, 60, 90, and 120 min, and serum insulin values at 0 min. In neonates, we registered sex, birth weight, height, Capurro index, and Apgar score at delivery. According to sex and gestational age, we calculated the birth weight percentile in all the neonates enrolled in the study using the Centers for Disease Control and Prevention (CDC) growth charts [18].

### 2.3. Design of Light or Heavy Sucralose-Consuming Groups

We invited pregnant women to participate in the study after estimating sucralose in-take frequency using the Food Frequency Questionnaire with Intense Sweeteners (FFQIS) previously validated in the Hispanic population [19]. We confirmed sucralose concentration in mothers’ serum and umbilical cord blood by high-performance liquid chromatography (HPLC). Based on previous studies [13,20], we grouped mothers into two groups according to the intensity of sucralose ingestion during pregnancy. Light sucralose-consuming mothers ate or drank less than 60 mg sucralose per week, equivalent to ingesting less than five commercial Splenda^®^ packets per week during pregnancy, representing less than 20% of the Acceptable Daily Intake (ADI) set by the Food and Drug Administration (FDA). Heavy sucralose-consuming mothers ate or drank more than 36 mg of sucralose per day, equivalent to ingesting more than three commercial Splenda^®^ packets daily during gestation. We estimated the sample size based on the results from Azad et al., expecting an effect size of 0.39 with an alpha error of 0.05 and a power of 95% for a difference between two independent means that resulted in a sample size of 286 participants [13].

### 2.4. Umbilical Cord Blood Samples

We collected umbilical cord blood samples of around 5–8 mL in all the neonates enrolled in the study by puncturing the umbilical vein 5 min after birth. Then, we equally divided the blood into a purple cap tube and a golden cap tube (Vacutainer, BD Diagnostics, NJ, USA) for the posterior isolation of white blood cells (WBC) and serum by centrifugation at 1800 g for 15 min at room temperature. After measuring glucose and insulin levels by the glucose oxidase assay (Megazyme International, Wicklow, Ireland) and the enzyme-linked immunosorbent assay (ELISA) (Abnova Corporation, Taipei, Taiwan), respectively, we properly stored WBC or serum samples until use.

### 2.5. Immunostaining and Flow Cytometry for Monocyte Subsets

Immediately after collecting WBC, we rinsed the cell pellet with 300 μL PBS 1X (Sigma Aldrich, St. Louis, MO, USA) and centrifuged at 1800× *g* for 15 min at room temperature. After adding 5 mL ammonium-chloride-potassium (ACK) lysing buffer (Thermo Fisher Scientific, Vienna, Austria), we resuspended the cell pellet gently and incubated it for 5 min at room temperature. Then, we centrifuged at 1800× *g* for 10 min and discarded the supernatant, rinsing and resuspending the cell pellet twice with 300 μL PBS 1X. Next, we resuspended 4 × 10^6^ WBC in 50 μL cell staining buffer (BioLegend, Inc., San Diego, CA, USA), adding 5 μL True-Stain Monocyte Blocker^TM^ (BioLegend, Inc., San Diego, CA, USA) for 10 min on ice. Immediately after, we added anti-CD14 PE/Cy7, anti-CD16 PE/Cy5, Zombie UVTM dye (BioLegend, Inc., San Diego, CA, USA), and anti-HLA-DR BUV661 (BD Biosciences, San Jose, CA, USA) for 20 min in darkness at 4 °C. After centrifuging and rinsing with cell-staining buffer, we added 150 μL PBS 1X, acquiring 10,000 cell events per test in triplicate corresponding to the HLA-DR+ cell population on a BD Influx flow cytometer (BD Biosciences, San Jose, CA, USA), using the BD software^TM^ version 1.2. (BD Biosciences, San Jose, CA, USA).

### 2.6. Gating Strategy for Monocyte Subsets

First, we gated WBC on a time/side scatter density plot, selecting the Zombie UV negative cell population as living cells. Then, we gated living cells for singlets on a forward scatter (FS)/Trigger Pulse Width density plot, selecting the HLA-DR+ cell population as monocytes. After using the rectangular gating method on the cell population expressing CD14 and CD16, we recognized classical monocytes (CM) as CD14++CD16- cells, intermediate monocytes (IM) as CD14++CD16+ cells, and nonclassical monocytes (NCM) as CD14+CD16+. We analyzed data with the FlowJo 10.0.7 software (TreeStar, Inc., Ashland, OR, USA).

### 2.7. Umbilical Cord Specimens for IL-1 Beta, TNF-Alpha, and IL-10 mRNA Expression

We collected 0.5 g umbilical cord tissue from all the neonates enrolled in the study 5 min after birth. Immediately after, we placed umbilical cord samples in TRIzol reagent (Thermo Fisher Scientific, Vienna, Austria) for the posterior isolation of total ribonucleic acid (RNA) using the phenol/chloroform/guanidine isothiocyanate method. After quantifying RNA by UV spectrophotometry, we generated complementary desoxyribonucleic acid (cDNA) using the M-MLV Retrotranscriptase system with dT primer (Thermo Fisher Scientific, Vienna, Austria) at 37 °C for 60 min. Then, we used cDNA for amplifying IL-1 beta, TNF-alpha, and IL-10 by the real-time quantitative polymerase chain reaction (qPCR) using SYBR Green Master Mix and AmpliTaq^®^ Fast DNA Polymerase (Thermo Fisher Scientific, Vienna, Austria) in the presence of specific primers. We utilized the 18S-ribosomal RNA sequence as house-keeping gene control, normalizing the expression of IL-1 beta, TNF-alpha, and IL-10 with the house-keeping gene expression to report it as fold change.

### 2.8. Statistics

We divided numerical and categorical data from mothers and neonates into light sucralose-consuming mothers and heavy sucralose-consuming mothers during pregnancy. After estimating the normality of data by the Shapiro-Wilk test, we compared numerical variables by the unpaired Student’s *t*-test, expressing the values as mean ± standard deviation. We compared categorical variables by the Chi-squared test, presenting the values as absolute numbers. Differences in birth weight between neonates lightly or heavily exposed to sucralose during gestation were adjusted by confounding variables by multiple regression analysis using the terminal R 3.5.1. We considered differences significant when *p* < 0.05, using the GraphPad Prism 7 software.

## 3. Results

There were no significant differences between light and heavy sucralose-consuming groups for age, blood pressure, pregestational BMI, number of previous pregnancies, gestation age, and CTOG’s glucose values (Table 1). The serum insulin concentration tended to increase in the heavy sucralose-consuming group compared to light sucralose-consuming mothers. However, no significant differences were reached (13.2 ± 4.5 vs. 10.2 ± 4.2, *p* = 0.0618).

As expected, mothers who referred to consuming sucralose occasionally exhibited 3.1 ± 1.4 ng/mL of serum sucralose compared to mothers reporting to ingest sucralose heavily, who showed a significant 8-fold increase of around 25.4 ± 4.2 ng/mL (*p* < 0.0001) (Table 1). On the contrary, sucralose concentration in umbilical cord blood samples was undetectable by the same HPLC methodology. Moreover, although there were no apparent differences between light or heavy sucralose-consuming groups for the SCP type more frequently consumed, we observed a significant increase in the number of SCP ingested weekly (*p* < 0.0001) (Table 1). In infants, there were no apparent differences between neonates born from mothers who lightly consumed sucralose and newborns from mothers who heavily ingested sucralose for sex proportion, birth height, Capurro index, and Apgar score (Table 1). Conversely, infants born from mothers who heavily consumed sucralose displayed a significant increase in birth weight compared to that found in newborns from mothers who lightly ingested sucralose (3.2 ± 0.6 vs. 2.8 ± 0.1, *p* = 0.0005) (Table 1). As a matter of fact, we more often observed neonates with a birth weight above the 95th percentile in the heavy sucralose-consuming group compared to that in the light sucralose-consuming group (14.9% (*n* = 13) vs. 7.3% (*n* = 15), *p* = 0.0470) (Table 1). The difference in birth weight between newborns lightly or heavily exposed to sucralose during gestation was maintained after adjusting by confounding variables, such as the mother’s pregestational BMI and the number of previous pregnancies (Table 2).

Weight gain is often associated with metabolic disturbances involving the circulating levels of glucose and insulin. In this sense, there were no differences between neonates born from mothers who lightly consumed sucralose during gestation and newborns from mothers who heavily ingested sucralose during pregnancy for glucose blood levels in the umbilical cord (*p* = 0.1719) (Figure 1A). Conversely, infants born from mothers who heavily consumed sucralose during pregnancy showed a significant increase in insulin levels compared to that found in neonates from mothers who lightly ingested sucralose (15.4 ± 5.7 vs. 12.2 ± 3.8, *p* = 0.0425) (Figure 1B).

Weight gain and metabolic alterations are linked to low-grade systemic inflammation, where monocyte subpopulations play a pivotal role. Figure 2 illustrates the gating strategy for detecting the three subsets of human monocytes, which vary apparently between neonates differently exposed to sucralose during gestation (Figure 2).

There were no differences between newborns from mothers in the light sucralose-consuming group and infants born from mothers in the heavy sucralose-consuming group for classical and intermediate monocyte subsets (*p* = 0.3410 and *p* = 0.2103, respectively) (Figure 3A,B). In contrast, the nonclassical monocyte percentage significantly increased in newborns from mothers in the heavy sucralose-consuming group compared to infants born from mothers in the light sucralose-consuming group (8.1 ± 0.7 vs. 4.9 ± 1.0, *p* < 0.001) (Figure 3C).

Besides alterations in the monocyte subset balance, the expression of IL-1 beta, TNF-alpha, and IL-10 are differential markers of low-grade systemic inflammation. Notably, the umbilical cord tissue of neonates born from mothers who heavily ingested sucralose during pregnancy exhibited a significant 3-fold increase in IL-1 beta expression compared to that found in newborns from mothers who lightly consumed sucralose (2.7 ± 0.5 vs. 1.0 ± 0.0, *p* = 0.003) (Figure 4A). TNF-alpha expression followed a similar behavior, significantly increasing in the umbilical cord specimens of infants from mothers who heavily ingested sucralose compared to that found in neonates from mothers who lightly consumed sucralose during gestation (1.2 ± 0.1 vs. 1.0 ± 0.0, *p* = 0.0488) (Figure 4B). Conversely, umbilical cord samples of newborns from mothers who heavily ingested sucralose during pregnancy showed a significant 5-fold decrease in IL-10 expression compared to that found in infants from mothers who lightly consumed sucralose (0.2 ± 0.06 vs. 1.0 ± 0.0, *p* < 0.001) (Figure 4C).

## 4. Discussion

A growing body of evidence strongly suggests that sucralose intake during pregnancy may be a risk factor for the offspring to develop obesity-related disorders, including weight gain, metabolic dysfunction, and systemic inflammation early in life [10,14,21,22]. Our results indicate that mothers who heavily ingested sucralose during gestation delivered heavier babies more often found above the 95th percentile of birth weight than mothers occasionally consuming this NNS. Similarly, Azad et al. observed that children born from mothers who frequently drank beverages containing sucralose and other NNS during pregnancy exhibited a higher BMI at one year than kids whose mothers sporadically consumed NNS [23]. Furthermore, our data show no differences between light or heavy sucralose-consuming groups for the kind of SCP more often ingested, where yogurt, diet sodas, candy bars, baked goods, jams, and gelatin are usually consumed by Mexican pregnant women. In contrast, we found that the main difference between light or heavy sucralose-consuming pregnant women is given by the amount of SCP eaten or drunk per week, which aligns with previous works reporting the type of food or beverages more often consumed by school-age children in Mexico [24]. These findings indicate we should be aware of the kind and amount of foods and beverages offered to pregnant women and children to avoid the onset of metabolic and immune abnormalities later in life.

The fact that sucralose appears to exert the ability to upregulate genes involved in lipogenesis in vitro and in vivo may explain the effect of maternal exposure to sucralose on the progeny’s weight gain [13]. In this sense, sucralose increases the expression of fatty acid synthase (FAS) and fatty acid-binding protein 4 (FABP4) on the in vitro cultured 3T3L1 adipocytes and adipose tissue of mouse offspring born to sucralose-fed dams [13]. Likewise, recent evidence shows that sucralose feeding enhances FAS expression in the liver of mice with HFD-induced steatosis via the taste receptor type 1 member 3 (T1R3), a member of the sucralose’s receptor family mediating sweet taste perception [25,26]. Taking this information together, it is feasible that sucralose of maternal origin may directly promote lipogenesis and fat mass expansion in the offspring via T1R3; thus, partially explaining the weight gain observed in neonates exposed to this NNS during gestation. We believe these results should encourage other research teams to explore the probable mechanisms through which sucralose may program the onset of obesity early in life, which could help recommend new public policies regarding using NNS during pregnancy.

In addition to increasing birth weight, sucralose intake during gestation boosted insulin levels in newborns. A consistent phenomenon described in several studies is that exposure to sucralose stimulates insulin production and release [27,28,29]. In vitro stimulation of the MIN6 pancreatic beta-cell line and mouse islet cells with sucralose results in increased insulin secretion via T1R2/T1R3 heterodimerization-dependent cyclic adenosine monophosphate (cAMP) release [30]. This information allows us to suppose sucralose can directly interact with beta cells to induce insulin release by activating T1R2 and T1R3-dependent signaling pathways. In mice, sucralose supplementation for nine weeks induces a 2-fold increase in serum insulin compared to control animals drinking water [31]. Moreover, our working group and other research teams have consistently shown that short- or long-term ingestion of sucralose elevates serum insulin, indicating this NNS can also increase insulin production or release in human adults [28,32,33,34]. In line with this body of information, our data show that neonates born from mothers who heavily consumed sucralose during pregnancy exhibit increased insulin levels compared to newborns from mothers who occasionally ingested this NNS.

To the best of our knowledge, this is the first report providing new evidence that sucralose may go from the mother’s bloodstream through the placenta to the newborn’s bloodstream via vertical passing, reaching the fetal pancreas to induce either insulin synthesis or secretion. Halasa et al. previously found sucralose in amniotic fluid, but not the umbilical cord blood, suggesting transplacental passing of this NNS with potential effects on the progeny’s metabolic health [35]. To some extent, Halasa’s report resembles our inability to detect sucralose in cord blood. On the contrary, we informed considerable amounts of sucralose in the mother’s serum, which is in line with previous reports showing up to 15% of sucralose can be absorbed intestinally and pass into the bloodstream [36,37,38]. Once in the mother’s bloodstream, we speculate that sucralose may reach the amniotic fluid via transplacental passing and then, the neonate’s bloodstream to exert its actions on the fetal pancreas in charge of producing insulin. The possible transplacental passing of sucralose may have several implications for reproductive health and pregnancy, where metabolic abnormalities can be programmed and affect the offspring later in life. It is, thus, urgent to demonstrate scientifically the transplacental passing of sucralose, which would allow us to recommend women not to use NNS during pregnancy; above all, female patients with a T2D familial history involving a pancreatic compromise in insulin production.

Finally, both cellular and humoral markers of low-grade systemic inflammation accompanied the increase in birth weight and insulin observed in neonates exposed to sucralose in utero. In humans, circulating monocytes are divided into classical, intermediate, and nonclassical monocytes according to the CD14 and CD16 cell-surface expression [39]. While classical and intermediate monocyte subsets preferably adhere to endothelial tissue promoting immune cell migration, the nonclassical monocytes exert inflammatory functions by expressing cytokines such as IL-1 beta and TNF-alpha [40,41]. The molecular mechanisms contributing to polarizing the activities of monocyte subsets from cell adhesion and migration to inflammation largely depend on CD14 expression, which decreases in the nonclassical monocyte subpopulation [42]. CD14 synthesis is regulated by specificity protein 1 (SP1), a transcription factor mostly expressed during human embryogenesis with the ability to orchestrate cell differentiation and growth, organogenesis, and myeloid cell maturation such as monocytes [43,44]. Interestingly, the sucralose receptor T1R3 can regulate downstream the protein kinase B (AKT), which, in turn, promotes SP1 activation [45]. In this scenario, it is feasible that sucralose ingestion during pregnancy may favor SP1 activation in myeloid cells; thus, regulating CD14 production in monocytes and conversion of these cells into the nonclassical subtype that exerts inflammatory actions belonging to the low-grade systemic inflammation.

In line with the apparent inflammatory polarization of monocytes toward the nonclassical subgroup, we also observed abnormally high TNF-alpha and IL-1 beta circulating levels in newborns from mothers who heavily consumed sucralose during gestation. As mentioned, nonclassical monocytes mainly produce IL-1 beta, a cytokine with central roles in chronic and acute inflammatory responses, apoptosis, and obesity-related cardiovascular diseases such as atherosclerosis [46,47]. Nonclassical monocytes also primarily secrete TNF-alpha, a potent inflammatory inducer that can increase insulin resistance and insulinemia by dephosphorylating the insulin receptor substrate (IRS) and AKT via protein-tyrosine phosphatase 1B (PTP1B) activation [48,49,50]. Conversely, we also found that neonates regularly exposed to sucralose in utero displayed low systemic levels of IL-10, a cytokine with potent anti-inflammatory actions leading to immunosuppression and tissue repair [51,52]. IL-10 seems to counteract the low-grade systemic inflammation instigated by IL-1 beta and TNF-alpha in non-obese subjects, promoting weight loss and insulin sensitivity in both mice and humans [53,54]. Present results expand on the body of evidence indicating that sucralose consumption during pregnancy can predispose the offspring to low-grade systemic inflammation, a condition concurring with weight gain and altered insulin secretion in patients with obesity and metabolic dysfunction. However, we should take this evidence with caution until establishing the generalizability of the current findings in other scenarios, as with the fact that pregnant women can be exposed to other NNS and not only sucralose or how long a woman has ingested sucralose before getting pregnant.

## 5. Conclusions

To the best of our knowledge, this is the first study providing evidence in humans that sucralose intake during pregnancy may predispose the neonate to weight gain, altered insulin secretion, and low-grade systemic inflammation early in life. Together with numerous studies in animal models, present results expand on the notion that sucralose and other NNS may act as obesogenic factors during fetal development, influencing the onset of obesity and metabolic disease in childhood. We encourage other research teams to conduct prospective cohort studies to follow up on newborns intrauterinally exposed to sucralose across the years to draw significant conclusions regarding the possible role of NNS in programming obesity and metabolic disease later in life. The urgent need for additional investigation in this field is justified when considering that up to 50% of obese children become obese in adulthood with a 4-fold increased risk of developing chronic non-communicable diseases such as T2D [55,56].

The study’s limitations include that we conducted a cross-sectional study that restricts us from finding cause-and-effect relationships among the studied variables. Moreover, we used HPLC to measure sucralose instead of ultra-high performance liquid chromatography coupled to mass spectroscopy, which probably would have improved our ability to detect sucralose in cord blood, even in tiny quantities. Conversely, the sample size is big enough to conclude that maternal sucralose intake during pregnancy is associated with increased birth weight, circulating insulin, and low-grade systemic inflammation in neonates born at term, a finding that demands further research. However, we think increasing the sample size would allow us to analyze data by quartiles, where we can compare the effect of never, occasionally, moderately, or heavily consuming sucralose on crucial neonatal parameters such as birth weight. Present results provide scientific evidence of the impact of maternal sucralose ingestion during gestation on the progeny that may help dictate new policies aimed at regulating NNS intake in pregnant women, above all, those with a familial history of obesity and T2D.

## Figures and Tables

**Figure 1 biomedicines-11-00650-f001:**
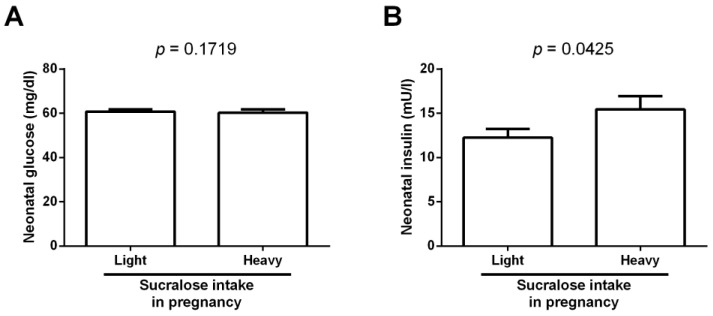
Glucose and insulin levels in neonates lightly or heavily exposed to sucralose during pregnancy. (**A**) Cord blood glucose levels in newborns lightly or heavily exposed to sucralose. (**B**) Cord blood insulin levels in neonates lightly or heavily exposed to sucralose. We measured glucose and insulin in umbilical cord blood from all infants. We expressed data as mean ± standard deviation. We compared data by the unpaired Student’s *t*-test, considering differences significant when *p* < 0.05.

**Figure 2 biomedicines-11-00650-f002:**
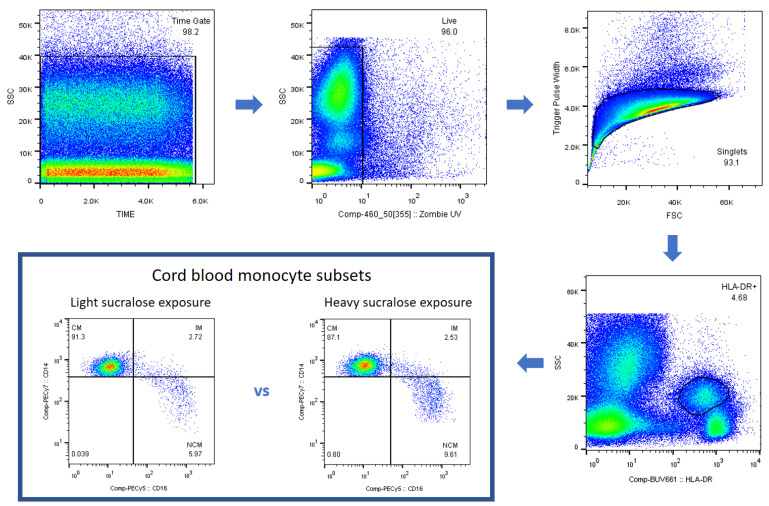
Gating strategy for detecting classical, intermediate, and nonclassical monocyte subsets in neonates lightly or heavily exposed to sucralose during pregnancy. CD14++CD16- cells were recognized as classical monocytes, CD14++CD16+ cells were identified as intermediate monocytes, and CD14+CD16+ cells were distinguished as nonclassical monocytes. The last panel illustrates how nonclassical monocytes significantly increase in neonates from mothers who intensely consumed sucralose during pregnancy. We acquired 1 × 10^4^ cell events per test in triplicate corresponding to the HLA-DR+ cell population on a BD influx flow cytometer.

**Figure 3 biomedicines-11-00650-f003:**
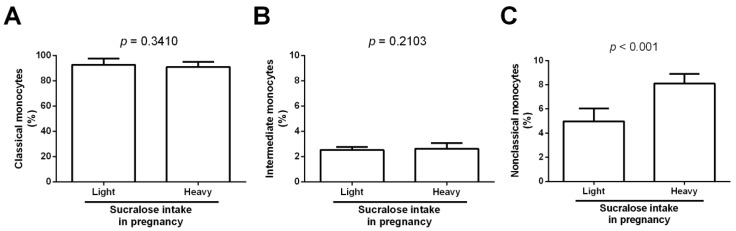
Percentages of classical, intermediate, and nonclassical monocytes in neonates lightly or heavily exposed to sucralose during pregnancy. (**A**) Classical monocyte percentage in newborns lightly or heavily exposed to sucralose. (**B**) Intermediate monocyte percentage in neonates lightly or heavily exposed to sucralose. (**C**) Nonclassical monocyte percentage in newborns lightly or heavily exposed to sucralose. We measured percentages of monocyte subsets in umbilical cord blood from all infants. We expressed data as mean ± standard deviation. We compared data by the unpaired Student’s *t*-test, considering differences significant when *p* < 0.05.

**Figure 4 biomedicines-11-00650-f004:**
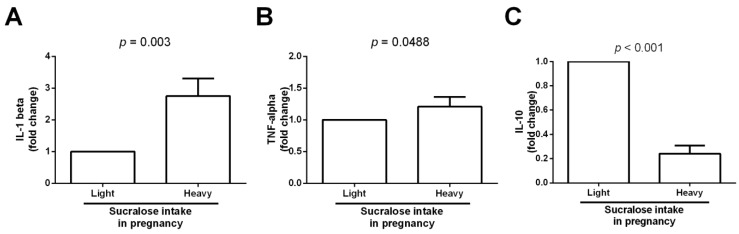
IL-1 beta, TNF-alpha, and IL-10 expression in umbilical cord specimens from neonates lightly or heavily exposed to sucralose during pregnancy. (**A**) IL-1 beta expression in umbilical cord tissue from newborns lightly or heavily exposed to sucralose. (**B**) TNF-alpha expression in umbilical cord samples from neonates lightly or heavily exposed to sucralose. (**C**) IL-10 expression in umbilical cord specimens from infants lightly or heavily exposed to sucralose. We expressed data as mean ± standard deviation. We compared data by the unpaired Student’s *t*-test, considering differences significant when *p* < 0.05. IL-1 beta, interleukin 1 beta; TNF-alpha, tumor necrosis factor alpha; IL-10, interleukin 10.

**Table 1 biomedicines-11-00650-t001:** Demographic and metabolic characteristics of the study participants.

		Sucralose Intake in Pregnancy		
	Characteristics	Light	Heavy	*p* Value
Mothers	Number of participants	205	87	-
	Age (years)	25.9 ± 5.3	26.1 ± 3.3	0.4675
	SBP (mmHg)	106.7 ± 11.5	106.9 ± 10.3	0.4869
	DBP (mmHg)	70.4 ± 9.7	70.6 ± 9.0	0.4827
	Pregestational BMI (kg/m^2^)	27.2 ± 4.3	27.4 ± 4.6	0.4591
	Previous pregnancies	1.6 ± 0.8	1.7 ± 0.8	0.3945
	Age of pregnancy (weeks)	39.1 ± 1.2	39.2 ± 1.3	0.4411
	OGTT blood levels at 0′ (mg/dL)	89.9 ± 6.1	90.9 ± 5.6	0.3545
	OGTT blood levels at 30′ (mg/dL)	122.3 ± 10.2	123.5 ± 9.3	0.3936
	OGTT blood levels at 60′ (mg/dL)	101.6 ± 31.1	99.0 ± 17.3	0.4101
	OGTT blood levels at 90′ (mg/dL)	93.0 ± 12.2	94.2 ± 13.7	0.4196
	OGTT blood levels at 120′ (mg/dL)	97.4 ± 14.6	97.1 ± 14.3	0.4818
	Serum insulin concentration (mU/L)	10.2 ± 4.2	13.2 ± 4.5	0.0618
	Serum sucralose concentration (ng/mL)	3.1 ± 1.4	25.4 ± 4.2	<0.0001 *
	Number of SCP eaten or drunk per week	2.0 ± 1.6	23.3 ± 1.5	<0.0001 *
	Type of SCP more often consumed	Yogurt, diet sodas, candy, baked goods, gelatin	Yogurt, diet sodas, candy, baked goods, jams, gelatin	-
Neonates	Sex (f/m)	109/96	48/39	0.7981
	Birth weight (kg)	2.8 ± 0.1	3.2 ± 0.6	0.0005 *
	Neonates above the 95th percentile (*n*)	15	13	0.0470 *
	Height (cm)	48.6 ± 2.4	50.1 ± 2.0	0.1044
	Capurro index (weeks)	39.3 ± 1.6	39.3 ± 1.5	0.9326
	Apgar score	8.9 ± 0.3	8.7 ± 0.4	0.2221

We show demographic, clinical, and biochemical data in mothers and neonates lightly or heavily exposed to sucralose during gestation. According to sex and gestational age, we calculated the birth weight percentile in neonates, using the CDC growth charts. We expressed data as mean ± standard deviation. We compared data by the unpaired Student’s *t*-test or the Chi-square test, considering differences significant when *p* < 0.05. Asterisks (*) indicate significant differences. SBP, systolic blood pressure; DBP, diastolic blood pressure; BMI, body mass index; OGTT, oral glucose tolerance test; CDC, Centers for Disease Control and Prevention; SCP, sucralose-containing products.

**Table 2 biomedicines-11-00650-t002:** Multiple linear regression analysis to estimate the influence of maternal variables in neonates’ birth weight.

Maternal Variables	β	SE	*T*	*p* Value
Age	−0.165	1.124	−0.132	0.880
SBP	0.878	0.679	1.030	0.772
DBP	0.896	1.826	0.636	0.523
Pregestational BMI	1.857	3.751	0.347	0.712
Previous pregnancies	−0.647	0.902	−0.651	0.503
Age of pregnancy (weeks)	0.023	0.018	0.347	0.764
Serum insulin concentration (mU/L)	8.250	3.720	2.328	0.061
Serum sucralose concentration (ng/mL)	0.376	0.571	0.714	0.520

Maternal variables did not significantly influence birth weight in neonates lightly or heavily exposed to sucralose during gestation. Multiple R-squared = 0.867, adjusted R-squared = 0.189, confidence interval = 2.5–97.5%, F-statistic = 1.218 on 33 and 6 degrees of freedom, *p* value = 0.403. We considered significant effects when *p* < 0.05. β, beta value; SE, standard error; *T*, *t* value; SBP, systolic blood pressure; DBP, diastolic blood pressure; BMI, body mass index.

## Data Availability

Data are available upon request.

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
