# Peer review of "Newborns from Mothers Who Intensely Consumed Sucralose during Pregnancy Are Heavier and Exhibit Markers of Metabolic Alteration and Low-Grade Systemic Inflammation: A Cross-Sectional, Prospective Study"

_biomedicines, 2023, doi:10.3390/biomedicines11030650_

Round 1

Reviewer 1 Report

Dear authors, I consider the paper to be very interesting and suitable for publication. The objectives of the article are very clear and the conclusions correspond to the objectives. The literature review is complete and up to date

I would only consider putting the p in small letters instead of capital letters (P=0.04).

Author Response

REVIEWER 1

Query (Q) 1. Dear authors, I consider the paper to be very interesting and suitable for publication. The objectives of the article are very clear and the conclusions correspond to the objectives. The literature review is complete and up to date.

Reply (R) 1. We appreciate your kind comments on our work.

Q2. I would only consider putting the p in small letters instead of capital letters (P=0.04).

R2. We replaced all uppercase P letters with lowercase p letters, as suggested. Please find these changes marked grey on pages 5, 6, 7, 8, 9, and 10, Table 1, Figure 1, Figure 3, and Figure 4. Thank you for your comments that have indubitably improved the last version of this manuscript.

Reviewer 2 Report

In the manuscript titled "Newborns from mothers who consumed sucralose intensely during pregnancy are heavier and exhibit marker of metabolic alteration and low-grade systemic inflammation," by José Alfredo Aguayo-Guerrero and colleagues. In comparison with newborns from LSI mothers, infants from HSI mothers displayed significant increases in birth weight and insulin. There was a significant increase in the percentage of inflammatory nonclassical monocytes in neonates from HSI mothers compared to neonates from LSI mothers. The umbilical cord tissue of infants born to HSI mothers exhibited higher levels of IL-1 beta and TNF-alpha, and lower levels of IL-10 than that of babies born to LSI mothers. Present results demonstrate that heavy sucralose ingestion during pregnancy affects neonates' anthropometric, metabolic, and inflammatory features. Regarding the present manuscript, I have a few comments.

-The topic is very interesting, but the authors have provided only a brief introduction. It may be possible to resolve this issue by adding updated information. There are several key aspects missing from this report, including public health data, overweight and obesity cases associated with sucralose consumption, or even prevention costs.

-I would like to thank you for giving me the opportunity to revise such a well-structured and well-written essay. I have provided some minor comments to the authors. The p-value is in lowercase, and no uppercase.

-The subsections 2.1 and 2.2 in the materials and methods section are missing. ...

-Why did they decide to measure sucralose when they planned to test another NNS?

-Is there a planned next step in the present study by the authors?

-By using other mathematical methods, the authors could enrich their data and perhaps replicate the results here in other populations from different locations or even countries

Author Response

REVIEWER 2

Query (Q) 1. In the manuscript titled "Newborns from mothers who consumed sucralose intensely during pregnancy are heavier and exhibit marker of metabolic alteration and low-grade systemic inflammation," by José Alfredo Aguayo-Guerrero and colleagues. In comparison with newborns from LSI mothers, infants from HSI mothers displayed significant increases in birth weight and insulin. There was a significant increase in the percentage of inflammatory nonclassical monocytes in neonates from HSI mothers compared to neonates from LSI mothers. The umbilical cord tissue of infants born to HSI mothers exhibited higher levels of IL-1 beta and TNF-alpha, and lower levels of IL-10 than that of babies born to LSI mothers. Present results demonstrate that heavy sucralose ingestion during pregnancy affects neonates' anthropometric, metabolic, and inflammatory features. Regarding the present manuscript, I have a few comments.

Reply (R) 1. Thank you for your comments on this work.

Q2. The topic is very interesting, but the authors have provided only a brief introduction. It may be possible to resolve this issue by adding updated information. There are several key aspects missing from this report, including public health data, overweight and obesity cases associated with sucralose consumption, or even prevention costs.

R2. We added a short paragraph where we updated information regarding the global increase in obesity and non-nutritive sweetener consumption, especially sucralose. We thank you for your observation, which has strengthened the introduction section. Please see this addition marked yellow on page 2.

Q3. I would like to thank you for giving me the opportunity to revise such a well-structured and well-written essay. I have provided some minor comments to the authors. The p-value is in lowercase, and no uppercase.

R3. We appreciate your kind comments on this work. We replaced all uppercase P letters with lowercase p letters, as suggested. Please find these changes marked grey on pages 5, 6, 7, 8, 9, and 10, Table 1, Figure 1, Figure 3, and Figure 4.

Q4. The subsections 2.1 and 2.2 in the materials and methods section are missing. Why did they decide to measure sucralose when they planned to test another NNS?

R4. With all due respect, we want to remark that we showed subsections 2.1 and 2.2 under the subheadings “Participants” and “Demographic, anthropometric, and clinical measurements” (pages 2 and 3, respectively). Furthermore, the results presented here belong to a more extensive research project aimed at examining the effects of the most consumed non-nutritive sweeteners on the metabolism and immunity of the Mexican population. In this first manuscript, we focused on sucralose because our data indicate it is one of the most ingested non-nutritive sweeteners among pregnant women. Marked green on page 2, you can find the rationale behind studying sucralose and no other non-nutritive sweetener in this first report.

Q5. Is there a planned next step in the present study by the authors?

R5. Yes. We will expand our findings by increasing the sample size and studying whether other non-nutritive sweeteners, such as aspartame or stevia, exert similar effects on human health. However, these data are part of a more extensive research project that will take three-to-four years at least to analyze and publish.

Q6. By using other mathematical methods, the authors could enrich their data and perhaps replicate the results here in other populations from different locations or even countries.

R6. We appreciate your accurate suggestion. We will incorporate mathematicians and statisticians into our study group to enrich these data and conduct a multicenter trial, including several countries, hopefully. We thank you for your comments and suggestions that indubitably improved the last version of the manuscript.

Reviewer 3 Report

Manuscript ID: biomedicines-2207395

Title: Newborns from mothers who intensely consumed sucralose during pregnancy are heavier and exhibit markers of metabolic alteration and low-grade systemic inflammation

The aim of this study was to evaluate the effects of sucralose ingestion during pregnancy on markers of metabolic dysfunction and systemic inflammation, including birth weight, glucose and insulin levels, monocyte subsets, and IL-1 beta, TNF-alpha, and IL-10 expression in a cohort of neonates born at term.

Comments and Suggestions for Authors:

The manuscript is an interesting study, but requires some considerations.

1. Title and Abstract:

The type of study design should be indicated.

2. Material and methods:

It is not shown how the calculation of the sample size necessary to obtain conclusions about the stated objectives was carried out. This is important to present.

It is also not clear the sampling strategy to include pregnant women in the study and why there were two sampling periods.

Line 112. Where it says: "We performed oral glucose tolerance tests (OGTT) in all pregnant women at the beginning of the third trimester of pregnancy, starting with a 2 g glucose load at 0 min". Why was a 2 g glucose load used? What type of OGTT was used? Please clarify this issue.

Line 118. The Centers for Disease Control and Prevention (CDC) growth charts bibliographic reference should be included.

Line 119. Design of light or heavy sucralose-consuming groups.

Why were these cut-off points chosen to classify between Light and Hight sucralose-consuming mothers? Based on what recommendation was made?

Line 120. Where it says: "We estimated sucralose intake during pregnancy using the Food Frequency Questionnaire with Intense Sweeteners (FFQIS) previously validated in the Hispanic population." These results are not presented and would be very interesting.

Line 183. It is indicated that "Differences between neonates slightly or heavily exposed to sucralose during gestation were adjusted by confounding variables by multiple regression analysis". However, these data are not shown in results.

3. Results:

In the Results section there is a tendency to repeat aspects that should already be specified in the Material and methods section, especially in the footers of all Tables and Figures. The footnotes of Tables and Figures should only be used to clarify the abbreviations used or some necessary aspect not previously described. Likewise, in the text it could be avoided to repeat data that appear in the section of Material and methods or in the Tables and it would be better to summarize them.

Indicate the number of participants with missing data for each variable considered.

4. Discussion:

It should be discussed and reconsidered if it would not have been better to change the design of light or heavy sucralose-consuming groups. Wouldn't it have been preferable to make a comparison by quartiles?

The generalisability (external validity) of the study results should be discussed.

The authors honestly acknowledge some limitations of the study. The calculation of the necessary sample size is not indicated.

Author Response

REVIEWER 3

Query (Q) 1. The aim of this study was to evaluate the effects of sucralose ingestion during pregnancy on markers of metabolic dysfunction and systemic inflammation, including birth weight, glucose and insulin levels, monocyte subsets, and IL-1 beta, TNF-alpha, and IL-10 expression in a cohort of neonates born at term. The manuscript is an interesting study, but requires some considerations.

Reply (R) 1. We appreciate your kind comments on this work.

Q2. Title and Abstract: The type of study design should be indicated.

R2. We added the study type to the title and abstract as requested. Please find these changes marked yellow on page 1.

Q3. Material and methods: It is not shown how the calculation of the sample size necessary to obtain conclusions about the stated objectives was carried out. This is important to present.

R3. Following your accurate observation, we added information on the sample size calculation. Based on the results from Azad et al. (Azad MB, et al., Nonnutritive sweetener consumption during pregnancy, adiposity, and adipocyte differentiation in offspring: evidence from humans, mice, and cells. Int J Obes (Lond). 2020;44(10):2137-2148), expecting an effect size of 0.39 with an alpha error of 0.05 and a power of 95% for a difference between two independent means resulted in a sample size of 286 participants. Please see this addition marked yellow on page 3.

Q4. It is also not clear the sampling strategy to include pregnant women in the study and why there were two sampling periods.

R4. We invited to participate in the study to pregnant women who voluntarily wanted to initiate prenatal care at the beginning of the second trimester of pregnancy, excluding pregnant women with more than 14 weeks of pregnancy at the time of the study’s start. This way, we could monitor the food and beverages they consumed from week thirteen of gestation to delivery. All enrolled women were between 18 and 30 years and had a low-risk pregnancy. Once enrolled, all participant women voluntarily attended the Department of Gynecology of the General Hospital of Mexico, where we started to register the amount and frequency of food and beverages they used to consume daily, weekly, and monthly. In this way, we identified light and heavy sucralose-consuming women to form the study groups. We followed up with all participants until delivery in the Department of Gynecology’s facilities, where we registered anthropometric variables and took blood samples from neonates. You can find this information in lines 90-106.

Additionally, we rephrased critical aspects of the enrollment strategy for pregnant women to avoid misunderstandings and explain better how we enrolled participants in the study. Please see these changes marked turquoise on pages 2 and 3.

We enrolled participants in two periods because of the COVID-19 pandemic’s beginning, a time when the hospital converted from a tertiary care center to a dedicated COVID-19 hospital, forcing us to involuntarily stop the study and resume it in the second semester of 2021.

Q5. Line 112. Where it says: "We performed oral glucose tolerance tests (OGTT) in all pregnant women at the beginning of the third trimester of pregnancy, starting with a 2 g glucose load at 0 min". Why was a 2 g glucose load used? What type of OGTT was used? Please clarify this issue.

R5. We mistyped 2 g instead of 75 g glucose because of a terrible mistake. We appreciate your accurate observation that helped us correct such a mistake. Please find this correction marked pink on page 3.

Q6. Line 118. The Centers for Disease Control and Prevention (CDC) growth charts bibliographic reference should be included.

R6. We added the missing reference as requested. Please see this change marked grey on page 3.

Q7. Line 119. Design of light or heavy sucralose-consuming groups. Why were these cut-off points chosen to classify between Light and Hight sucralose-consuming mothers? Based on what recommendation was made?

R7. We classified light or heavy sucralose-consuming groups based on Maslova et al. 2013 and Azad et al. 2020, who estimated consumption of sucralose (among other NNS) in four groups: “never,” consuming less than a packet per month, “rarely,” consuming less than a packet per week, “lightly,” consuming two-to-six packets weekly, and “heavily,” consuming more than one packet daily. Notably, we found that pregnant women in our study fell into light and heavy groups, where almost 2 in 3 women used to ingest five sucralose packets per week (light group). In contrast, the rest of the participants eat or drink sucralose, equivalent to one-to-three packets daily. We believe your question is relevant and decided to add the references mentioned above. Please see this addition marked red on page 3.

Q8. Line 120. Where it says: "We estimated sucralose intake during pregnancy using the Food Frequency Questionnaire with Intense Sweeteners (FFQIS) previously validated in the Hispanic population." These results are not presented and would be very interesting.

R8. Following your suggestion, we added in Table 1 the results found when applied the FFQIS to study participants. The FFQIS is focused on exploring two main items, the number of sucralose-containing products (SCP) eaten or drunk per week and the type of SCP more often consumed. Although there were no apparent differences between light or heavy mother groups for the SCP type more frequently consumed, we observed a significant increase in the number of SCP ingested weekly. As you mentioned, we believe these results are exciting and added this information in the Material and Methods, Results, Discussion, and Table 1. Please find these additions marked green on pages 3, 5, 6, and 10.

Q9. Line 183. It is indicated that "Differences between neonates slightly or heavily exposed to sucralose during gestation were adjusted by confounding variables by multiple regression analysis". However, these data are not shown in results.

R9. Following your observation, we added a table denominated Table 2, showing a multiple regression model to estimate the potential influence of maternal variables in neonates’ birth weight. Please see these changes marked yellow on pages 4, 5, and 6.

Q10. Results: In the Results section there is a tendency to repeat aspects that should already be specified in the Material and methods section, especially in the footers of all Tables and Figures. The footnotes of Tables and Figures should only be used to clarify the abbreviations used or some necessary aspect not previously described. Likewise, in the text it could be avoided to repeat data that appear in the section of Material and methods or in the Tables and it would be better to summarize them.

R10. Following your recommendation, we deleted unnecessary sentences that repeated in the Result section what we already described in the Material and Method section. We also eliminated repeated information in the Tables and Figures’ footnotes. Please find these changes along with the Result section and Tables and Figures’ footnotes. We believe the Result section is more precise and understandable. Thank you for your observation.

Q11. Indicate the number of participants with missing data for each variable considered.

R11. We want to clarify that according to elimination criteria, we eliminated from the study participants with missing data, enrolling 292 mothers and neonates from whom we registered all demographic, anthropometric, clinical, and biochemical parameters. However, we think we had not explained this in the manuscript and decided to add a sentence to clarify it. Please see this addition marked blue on page 3.

Q12. Discussion: It should be discussed and reconsidered if it would not have been better to change the design of light or heavy sucralose-consuming groups. Wouldn't it have been preferable to make a comparison by quartiles?

R12. Following your recommendation, we reanalyzed data by quartiles of sucralose ingestion (we added a representative figure below showing neonatal insulin and nonclassical monocyte percentage as two crucial variables reported by light or heavy sucralose-consuming groups). However, we found no significant differences in neonatal insulin (A) or nonclassical monocyte percentage (B) when analyzing data by quartiles because the n substantially shortens in each group (the same applies to other parameters such as birth weight or cytokine expression pattern). Furthermore, when we analyzed the sucralose consumption pattern (both sucralose serum concentration (C) or number of sucralose-containing products ingested per week (D)) by quartiles we observed no differences among Q1, Q2, and Q3, finding significant changes only when comparing Q4 with the rest of quartiles. We think this is because participants in our study show a well-established sucralose consumption pattern either occasionally ingesting or heavily consuming this nonnutritive sweetener. For this reason, we still believe the comparison of neonatal variables in light or heavy sucralose-consuming groups is more informative by now. However, we have added a sentence discussing the possibility of analyzing data by quartiles in a further study with larger sample size. Please find this addition marked yellow on page 12.

Q13. The generalisability (external validity) of the study results should be discussed.

R13. Following your recommendation, we added a sentence discussing the generalisability of our findings in other settings, as with the fact that pregnant women can be exposed to other NNS and not only sucralose or how long a woman has ingested sucralose before getting pregnant. Please see this addition marked green on pages 11 and 12.

Q14. The authors honestly acknowledge some limitations of the study. The calculation of the necessary sample size is not indicated.

R14. We believe recognizing limitations to the study can help other research teams improve their study protocols. As mentioned above, we showed the sample size calculation that you can see marked yellow on page 3. We really appreciate all your comments, questions, and concerns, which have helped us correct mistakes and improve the last version of the manuscript.

Round 2

Reviewer 3 Report

In the new V2 manuscript, the authors have made changes based on the recommendation of referees that improve its presentation.